# Correlation between the tissue ghrelin presence, disease activity and laboratory parameters in ulcerative colitis patients; immunohistochemical study

**Memduh Sahin**[1]*, **Kivilcim Eren Erdogan**[2], **Emine Tekingündüz**[3]

**1** Department of Gastroenterology, Saglik Bilimleri University Sisli Hamidiye Etfal Training and Research Hospital, Istanbul, Turkey, **2** Department of Pathology, Cukurova University, Adana, Turkey, **3** Department of Pathology, Mersin City Hospital, Mersin, Turkey

* memsahinsahin@hotmail.com

## Abstract

**Data Availability Statement:** All relevant data are within the paper and its Supporting Information files.

### Background

The aim of the study was to determine the differences in terms of ghrelin presence in the colon between the patients with ulcerative colitis (UC) and control patients.

### Methods

Sixty-one UC and 15 control patients were included in the study. Immunohistochemical staining for ghrelin was investigated in colonic biopsy samples of UC and control patients. UC patients were subdivided into Group A (absence of ghrelin staining) and Group B (presence of staining for ghrelin in biopsy samples). Disease activity scores, laboratory parameters and quantitative ghrelin staining were compared in both groups of UC patients, as well as with the observations in control patients.

### Results

Cells in colonic mucosa stained for ghrelin were identified in twenty-three (37.7%) UC patients, while this proportion in control patients was 6/15(40%). A significant difference was found between Groups A and B for serum albumin concentration but not for *erythrocyte sedimentation rate* (ESR), C-reactive protein (CRP), hemoglobin concentration or leucocyte count. Mayo score/disease activity index (DAI) for UC were significantly higher in Group A than in Group B (p = 0.03).

### Conclusions

There were no differences in the amount of colonic ghrelin staining between healthy individuals and UC patients. Colonic ghrelin staining in UC patients seems to be associated with the increased activity of this disease.

**Funding:** The authors received no specific funding for this work.

**Competing interests:** The authors have declared that no competing interests exist.

## Introduction

Ulcerative colitis (UC) causes chronic inflammation of the colon and may have recurrence and remission cycles [1]. Pathogenesis of UC is multifactorial, including abnormal immune response, genetic predisposition, epithelial barrier defects and environmental factors [2]. Ghrelin, a peptide hormone containing 28 amino acids, is mainly produced by endocrine cells in gastric mucosa; in rats, these cells are called X/A-like cells, in humans they are called P/D1 cells [3]. Recent studies have shown that Ghrelin is involved in a number of gastrointestinal pathologies and immune system regulation [4, 5]. Apart from the stomach, ghrelin-producing cells and ghrelin mRNA were also found in the kidney, large intestine, rectum, small intestine, thyroid, placenta, brain, adrenal glands and ovaries, but the number of these cell and amount of ghrelin mRNA in these cells is much lower than in gastric mucosa [6–9]. Ghrelin is an endogenous ligand for ghrelin receptor [10] previously known as growth hormone secreta-gogue receptor 1a (GHS-R) [11]. Ghrelin receptor is mainly expressed in the pituitary gland and hypothalamus, but some level of ghrelin receptor is also found in other central and peripheral tissues [12, 13]. Ghrelin has been shown to increase the release of nitric oxide (NO) in the intestine, thus producing an antioxidant effect [14]. Ghrelin has also a protective role against stress-induced gastric ulcer [15]. NO production is associated with decreased tissue ulceration, congestion, vascular permeability and cellular infiltration. It has been shown that significant amounts of ghrelin are produced near the proliferative layer of the intestinal system, suggesting that ghrelin may have a direct role in enterocyte transformation or proliferation [16].

In patients with active inflammatory bowel disease, serum ghrelin concentrations are significantly elevated and positively correlated with serum inflammatory markers, such as tumor necrosis factor-α (TNF-α), C-reactive protein (CRP), erythrocyte sedimentation rate (ESR) and fibrinogen [17, 18].

The aim of this study was to determine the differences in tissue ghrelin expression, which may be indicative of an inflammatory effect, by comparing UC patients with a healthy control group. In addition, disease activity scores and laboratory parameters were compared between UC patients with immunohistological evidence of the presence or absence of ghrelin in colonic biopsy samples.

## Materials and methods

Individuals between the ages of 18–71, who had attended Mersin State Hospital between 2013 and 2016 and undergone colonoscopy, were included in the study. Sixty-one of the participants had been diagnosed with UC. The control group consisted of 15 individuals, who underwent colonoscopic biopsy for screening. The biopsies in control patients taken in the macroscopically healthy and unsuspected mucosa in the colon. Biopsy samples from UC patients were graded with Modified Riley Histopathological scoring method (MRS), which ranges from 0 (no inflammation) to 7 (severe acute inflammation) [19]. Mayo activity scores (DAI) [20], including stool frequency, rectal bleeding, endoscopic results, and physician's overall evaluation of all patients were calculated and documented. The UC group were divided into two sub-groups based on the presence or absence of staining for ghrelin on immunohistological examination of biopsy samples; patients, who stained positively for ghrelin were designated Group B and those without ghrelin staining were designated Group A.

The inclusion criteria were: all individuals, who underwent colonoscopy and who also consented to take part in the study. The exclusion criteria were: any patients diagnosed with diabetes; any patients with a body mass index $>30 \text{ kg/m}^2$; any patients with an oncological disease; any patients with kidney and/or liver failure; and any patients with advanced heart failure.

Blood Serum and CBC (Complet Blood Cell Count) samples were taken from the antecubital area of the subjects between 7 a.m. and 9 a.m., after 8 h fasting. Hemoglobin, ESR, White Blood Cell count, serum albumin and CRP values of all subjects (UC patients and control group) were analyzed. CBC samples installed in Ethylene Diamine Tetra Acetic Acid (EDTA) tubes. The Serum samples were collected in clean polypropylene tube (Blood centrifuged at 3000 rpm for 15 min).

Bowel biopsy specimens were fixed in buffered 10% formalin for 24 hours. Samples were embedded in paraffin wax, sectioned (4 μm thickness), and mounted on slides. Immunohistochemical staining of GPCs (Ghrelin positive cells) was performed using the streptavidin–biotin–peroxidase complex method (Thermo Fisher Scientific, USA). Paraffin-embedded, slide-mounted specimens were cleared of paraffin and dehydrated. To enhance the immunoreactivity of ghrelin, antigens were enhanced by treating with citric acid buffer at 95˚C for 40 minutes. After blocking endogenous peroxidase activity for 20 minutes with methanol containing 1% hydrogen peroxide ($H_2O_2$), the sections were bathed in normal goat serum for 15 minutes to prevent nonspecific binding. The sections were then incubated with rabbit anti-ghrelin polyclonal antibody (Abcam, Cambridge, UK) at 4˚C overnight. The next day, the sections were washed in 0.01 M phosphate-buffered saline (PBS) and incubated for 20 minutes with biotinylated goat anti-rabbit immunoglobulin G antibody (10 mg/ml). After rewashing with PBS, the sections were re-incubated for 20 minutes with peroxidase-conjugated streptavidin (100 mg/ml) and stained with 3,30-diaminobenzidine tetrahydrochloride in 0.05 M tris–HCl buffer containing $H_2O_2$. The sections were washed once more in PBS and counterstained with hematoxylin and eosin. Negative controls were treated identically but without the primary antibody.

Semiquantitative evaluation of the GPCs was performed by counting the number of positive cells in 10 microscope fields of mucosa at 400x magnification. The mean number of cells per field was calculated.

## Statistical analysis

All analyses were performed using SPSS version 16.0 for Windows (IBM Inc, Chicago, Illinois, USA). Kurtosis and skewness values between +2 and −2 were considered to indicate normal distribution and parameters outside these limits indicated non-normal distribution. Differences in ghrelin staining between UC patients and controls were assessed with the Mann-Whitney U test. In addition, the chi-square test and Fisher exact test were used for the analysis of categorical variables. A p-value of <0.05 was considered statistically significant. Power analysis was used to determine the effect of a given sample size with a given degree of confidence. The study was approved by the Ethics Committee of Mersin University (No:2016/28), and informed consent was obtained from all participants.

## Results

In twenty-one (34.4%) patients with UC, positive cells were detected with ghrelin dye, while in forty (65.6%) individuals, ghrelin staining was not detected. The majority of patients participating in the study were receiving oral mesalazine treatment (22 (36.1%) patients). Positive staining was observed in six (40%) individuals in the control group, whereas no staining was observed in the remaining nine (60%) individuals. There was no difference between the control and UC groups in the proportion of patients with ghrelin positivity (p = 0.92). Fig 1 shows a colon biopsy sample from a case of UC with cells positive for ghrelin shown stained brown. An example of a control sample stained for ghrelin is shown in Fig 2. Semi-quantitative assessment of ghrelin staining in biopsy samples from UC and control groups is shown in Table 1.

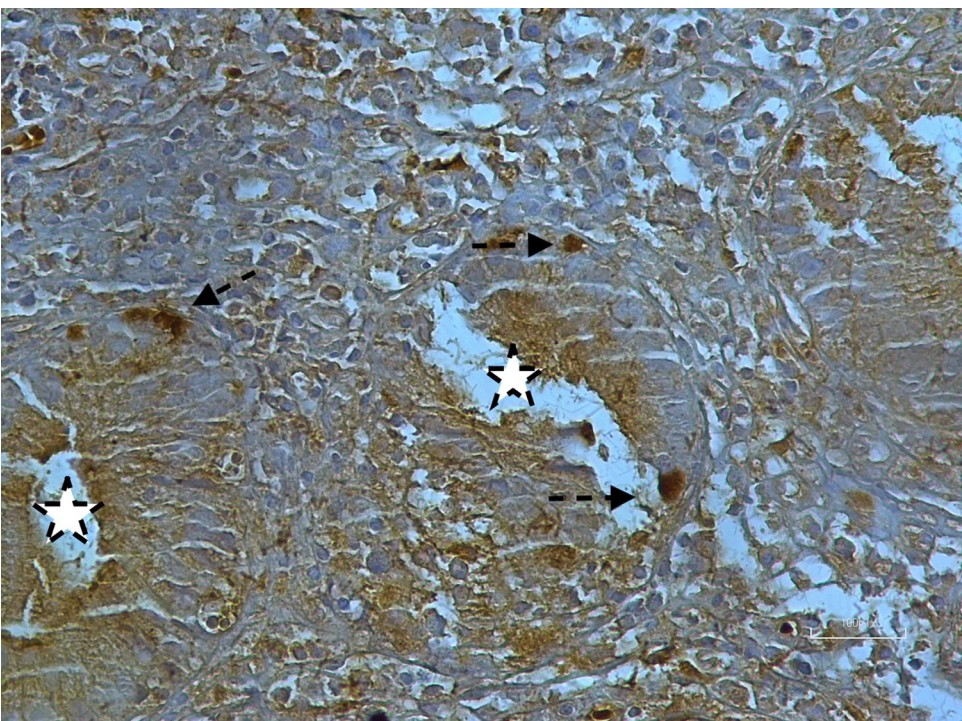

**Fig 1. Shows a colon biopsy sample from a case of UC with cells positive for ghrelin shown stained brown.**
Cryptitis (asterisks) in a case of inflammatory bowel disease. Immunohistochemical nuclear positivity in epithelial cells of crypts (arrows) (10 fields of mucosa at 400x magnification).

In histopathological scoring with MRS, there was no difference between group A (Median MRS: 7) and group B (Median MRS: 7) patients (p = 0.39). The mean DAI in Group A was 6.62 compared with 5.28 in Group B which was statistically significant (p = 0.03). The only laboratory parameter which was significantly different between Groups A and B was mean

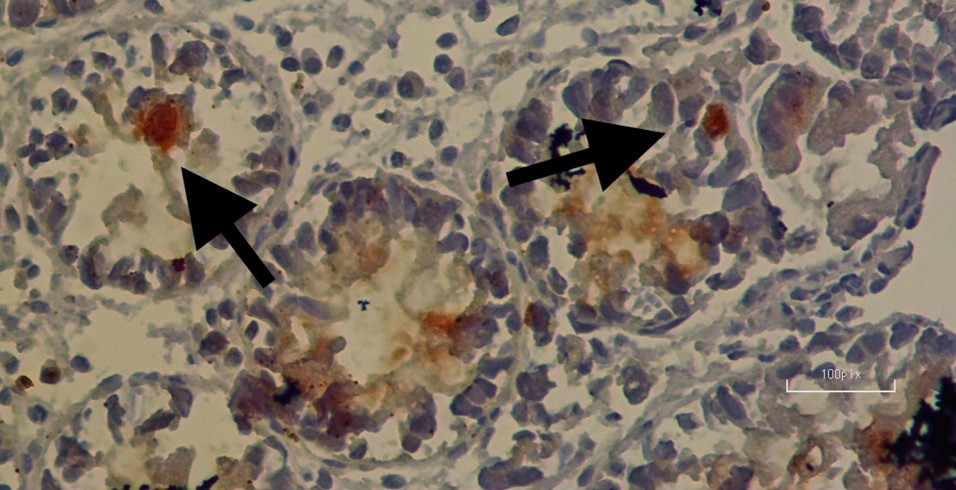

**Fig 2. An example of a control sample stained for ghrelin (Immunohistochemical nuclear positivity in epithelial cells of crypts (arrows)) (10 fields of mucosa at 400x magnification).**

**Table 1. Differences in ghrelin staining, biochemical parameters and age between control group and ulcerative colitis patients.**

|  | UC (n = 61) | Control (n = 15) | p |
|---|---|---|---|
| Amount of staining with Ghrelin *(min-max) | 0 (0–2) | 0 (0–15) | 0.91 |
| Alb (g/dl) ±SD | 4.18±0.31 | 4.27±0.36 | 0.04 |
| Age (Years) ±SD | 39.02±12.8 | 44.13±14.27 | 0.18 |
| Hb (g/dl) ±SD | 13.1±2.29 | 15.6±1.38 | 0.01 |
| WBC ($10^3$/mm$^3$)±SD | 8604±3.86 | 6559±1.22 | 0.048 |
| ESR ±SD | 19.74±16.07 | 11.14±2.99 | 0.01 |
| CRP (mg/L) | 1.11±1.45 | 0.32±1.11 | 0.01 |

* Median number of cells with positive staining in 10 fields

Mean±SD was given at regular intervals and median (min-max) at irregular intervals

Abbreviations: hsCRP–high sensitivity C-reactive protein; ESR–erythrocyte sedimentation rate; Hb–hemoglobin concentration; WBC–White blood cell count; Alb–Serum albumin concentration; SD–standard deviation.

albumin concentration (4.24±0.34 vs 4.07±0.18 g/dL, respectively; p = 0.04). There was no gender difference between Group A and Group B (p = 0.6). Comparison of laboratory and clinical parameters in Groups A and B is shown in Table 2.

While group A patients have the most common type of extensive colitis (42.8%), rectal involvement is the most common in group B patients (40%). The patients were examined in terms of the location of colonic lesions and the frequency of ghrelin positivity (see Table 3).

The majority of patients participating in the study were receiving oral mesalazine treatment (22 (36.1%) patients). Group A patients were most frequently treated for oral mesalazine, while Group B patients were most frequently treated for rectal meselazine. The treatments received by all patients are shown in Table 4.

Logistic regression analysis was used to examine the relationship between patient variables and the presence of ghrelin positive cells (see Table 5). Significant associations were identified for patient age (p = 0.047), C-reactive protein (p = 0.043) and Serum albumin concentration (p = 0.009) only.

**Table 2. Comparison of Mayo activity score (DAI), Modified Riley Score (MRS) and laboratory parameters in UC patients without (Group A) and with (Group B) ghrelin positivity.**

|  | Group A | Group B | p |
|---|---|---|---|
|  | n = 40 | n = 21 |  |
| CRP (mg/L) (min-max) | 1.15 (0.1–8) | 1.05 (0.1–6) | 0.81 |
| ESR ±SD | 19.11 ±14.2 | 20.80 ±19.3 | 0.69 |
| Hb (g/dl) ±SD | 13.2±2.25 | 12.97±2.42 | 0.72 |
| WBC ($10^3$/mm$^3$)±SD | 8.53 ± 3.73 | 8.76 ±4.20 | 0.84 |
| Age (Years) | 39.32±12.3 | 38.5±14.00 | 0.8 |
| Alb (g/dl) ±SD | 4.24 ±0.34 | 4.07 ±0.18 | 0.04 |
| DAI | 6.62±2.37 | 5.28±2.18 | 0.03 |
| Modified Riley Score (min-max) | 7 (3–7) | 7 (1–7) | 0.39 |

Abbreviations: hsCRP–high sensitivity C-reactive protein; ESR–erythrocyte sedimentation rate; Hb–hemoglobin concentration; WBC–leucocyte count; Alb–Serum albumin concentration; DAI–Mayo activity score; SD–standard deviation.

Mean±SD was given at regular intervals and median (min-max) at irregular intervals

**Table 3. Frequency analysis of ghrelin positivity according to colonic disease involvement area (ghrelin dye negative (Group A) and ghrelin dye-positive (Group B) UC patients).**

|  | Group A | Group B |
|---|---|---|
|  | n;(%) | n;(%) |
| Lesions were not evident, despite the diagnosis of UC | 3 (7.5%) | 1 (4.8%) |
| Rectum | 16 (40%) | 6 (28.6%) |
| Left colon type | 12 (30%) | 5 (23.8%) |
| Extensive type | 9 (22.5%) | 9 (42.8%) |
| Total | 40 (100%) | 21 (100%) |

**Table 4. Treatments used by ulcerative colitis patients (ghrelin dye negative (Group A) and ghrelin dye positive (Group B) UC patients).**

| Treatments | Group A | Group B | Total UC patients | PERCENT (%) |
|---|---|---|---|---|
|  | n = 40 | n = 21 | n = 61 |  |
| Meselasine Enema | 11 (27.5%) | 9 (42.9%) | 20 | 32.8% |
| Meselasine | 15 (37.5%) | 7 (33.3%) | 22 | 36.1% |
| Immunosuppressant treatment* | 10 (25%) | 5 (23.8%) | 15 | 24.6% |
| Without treatment | 4 (10%) | 0 | 4 | 6.6% |

* azathioprine, Anti-Tumor necrosis factor treatment, Steroid

**Table 5. Logistic regression analysis (last step of backward wald analysis) of the variables, associated with ghrelin positivity or negativity in biopsy samples from UC patients.**

|  | B | SE | t | p | OR (95% CI) | Adjusted R square |
|---|---|---|---|---|---|---|
| Dependent variable ghrelin |  |  |  |  |  | 0.49 |
| Crp | -8.41 | 0.416 | 4.08 | 0.043 | 0.191–0.975 |  |
| Sedimentation | -0.97 | 0.052 | 3.545 | 0.060 | 0.820–1.004 |  |
| Serum Albumin | -10.436 | 3.981 | 6.872 | 0.009 | 0.001–0.072 |  |
| Age | -0.097 | 0.049 | 3.949 | 0.047 | 0.825–0.999 |  |

*Dependent variable was Ghrelin negative (Group A) and positive (Group B) UC patients and independent variables in the regression model (Backward) were C-reactive protein, (CRP), leucocyte count, erythrocyte sedimentation rate (ESR), Serum albumin and hemoglobin concentrations, gender and patients age (year)

## Discussion

In this study, the presence of cells staining positive for ghrelin using immunohistochemical technique, which may be an indicator of inflammation, was compared between UC patients and controls. No difference was found between the UC and control groups in terms of immunohistochemical staining. Comparison between UC patients with and without ghrelin positivity showed serum albumin concentration to be significantly higher in the ghrelin negative group. In logistic regression models serum albumin, CRP concentration and patient age may be associated with positive staining for ghrelin in tissue samples taken from UC patients. DAI was higher in group A patients.

Previous studies have examined blood ghrelin concentrations in patients with inflammatory bowel conditions and have analyzed the relationship of these concentrations with other inflammatory markers. Ates, *et al.*, found no statistical differences in blood ghrelin concentration between healthy controls, UC and Crohn disease patients [17]. The same study found higher concentrations of blood ghrelin in patients with active disease in both the UC and Crohn patients. Similarly, Trejo-Vasquez, *et al.*, [21] found no difference in ghrelin

concentration between healthy control and UC patients [21]. Thus, it may be conjectured that blood ghrelin concentrations may be a marker of active disease rather than a marker of inflammatory bowel disease *per se*, or when the condition is quiescent.

In our study, higher levels of staining were found in the control group in terms of tissue staining amounts, but these data were not statistically significant. DAI were found to be higher in UC patients with showing ghrelin positivity (Group A). The relationship between disease activity and blood ghrelin concentrations in previous studies [17, 21] and DAI and tissue sample ghrelin positivity in our study appears to be similar.

Jung, *et al*., measured ghrelin mRNA concentrations in tissue biopsies of UC patients [22]. This research was conducted with twenty one patients (Twelve active and nine inactive patients attended in the research according to DAI. They reported that colon ghrelin mRNA concentrations were higher in patients with active disease compared to those in remission. Although we did not measure ghrelin mRNA concentration in our tissue samples, relying solely on histological examination, our cohort was larger than that of Jung, *et al.*

Studies have been conducted in models of colitis to examine the role of ghrelin. Konturek, *et al*., showed that exogenously administered ghrelin improved colitis (intrarectal trinitrobenze sulphonic acid-induced colitis models) in mice [23]. They suggested that increased NO levels and prostaglandin E2 release may have been responsible for pathological healing and increased colonic blood flow. In one study performed by Sahin, et al., [24], immunohistochemical ghrelin staining was performed in gastric biopsy samples in Irritable Bowel Syndrome (IBS) patients and the authors reported that higher intensity Ghrelin staining was found in constipation dominant IBS patients [24].

Kim, *et al*., [21] has examined the relationship between biochemical parameters and serum ghrelin concentrations in patients with Crohn's Disease [25]. Regression analysis revealed that CRP concentrations were positively associated with ghrelin, while there was a negative correlation between age and serum ghrelin concentration [25].

Our research is the first study of immunohistochemical staining of tissue samples taken from patients with UC and our results showed no difference in the proportion of individuals in the patient or control groups with ghrelin staining in cells. One of the limitations of our research was that blood ghrelin concentrations were not investigated simultaneously. Another limitation was the lack of tissue ghrelin mRNA concentration assessment in the samples examined immunohistochemically.

## Conclusion

There was no difference between the degree of ghrelin staining amounts from patients with UC and from control samples. However, higher serum albumin concentrations were detected in ghrelin-positive staining UC patients. Patient age, CRP and serum albumin concentration were found to be significantly associated with ghrelin expression in cells of colonic biopsies regardless of the type and position of lesion identified in the patients. Disease activity detected in the group with positive ghrelin staining was found to be statistically lower than in the negative stained group. However, since this was the first study in which ghrelin was investigated immunohistochemically in colon tissue biopsies from patients suffering from UC there is a need for further studies to confirm and expand on our results.

## Supporting information

**S1 File.**
(SAV)

## Acknowledgments

All authors contributed to the study and approved the final manuscript.

## Author Contributions

**Conceptualization:** Memduh Sahin.

**Data curation:** Memduh Sahin, Kivilcim Eren Erdogan, Emine Tekingündüz.

**Formal analysis:** Memduh Sahin, Kivilcim Eren Erdogan, Emine Tekingündüz.

**Investigation:** Memduh Sahin, Kivilcim Eren Erdogan, Emine Tekingündüz.

**Methodology:** Memduh Sahin, Kivilcim Eren Erdogan, Emine Tekingündüz.

**Project administration:** Kivilcim Eren Erdogan.

**Supervision:** Memduh Sahin.

**Validation:** Memduh Sahin.

**Visualization:** Memduh Sahin.

**Writing – original draft:** Memduh Sahin, Kivilcim Eren Erdogan.

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
