## [Decision Letter · Decision Letter 0]

29 May 2022

PONE-D-22-04274Relationship of tissue ghrelin levels with disease activity and laboratory parameters in ulcerative colitis patients; immunohistochemical studyPLOS ONE

Dear Dr. Sahin,

Thank you for submitting your manuscript to PLOS ONE. After careful consideration, we feel that it has merit but does not fully meet PLOS ONE’s publication criteria as it currently stands. Therefore, we invite you to submit a revised version of the manuscript that addresses the points raised during the review process. Please make sure, that in the abstract as well as in the main manuscript the same definitions and patient groups are addressed (see reviewer 1). One reviewer asks for the inclusion of a long list of additional literature. Please make sure that the cited literature adds to your introduction and discussion to avoid a lengthy literature list.Reviewer 2 asks why only UC and no CD patients have been included which might be an interesting question. Please add relevant therapies into your description of the patients. Please make sure that the discussion is based on your data and avoid speculations.

We look forward to receiving your revised manuscript.

Kind regards,

Gernot Zissel, Ph.D.

Academic Editor

PLOS ONE

Journal Requirements:

"No Fınancial Support"

b) State what role the funders took in the study. If the funders had no role in your study, please state: “The funders had no role in study design, data collection and analysis, decision to publish, or preparation of the manuscript

5. During your revisions, please note that a simple title correction is required: 'Relationship' is misspelled. Please ensure this is updated in the manuscript file and the online submission information

Reviewers' comments:

Reviewer's Responses to Questions

**Comments to the Author**

1. Is the manuscript technically sound, and do the data support the conclusions?

Reviewer #1: Partly

Reviewer #2: Partly

2. Has the statistical analysis been performed appropriately and rigorously? 

Reviewer #1: N/A

Reviewer #2: Yes

3. Have the authors made all data underlying the findings in their manuscript fully available?

Reviewer #1: Yes

Reviewer #2: Yes

4. Is the manuscript presented in an intelligible fashion and written in standard English?

Reviewer #1: No

Reviewer #2: Yes

5. Review Comments to the Author

Reviewer #1: Manuscript ID: AJGASTRO-D-21-00058

Title: Relationship between the tissue ghrelin presence, disease activity and laboratory parameters in ulcerative colitis patients; immunohistochemical study

Authors: Memduh Sahin et al.

The aim of above manuscript was to determine the relationship between the presence of ghrelin in the colon and the severity of ulcerative colitis. The topic of the study is interesting; however, it should be noted out that the manuscript contains many errors and shortcomings. For this reason, the manuscript must be corrected before possible acceptance for publication.

List of shortcomings and errors:

1. First of all, the authors check correctness of description of groups of people participating in the study and the presented results. In the abstract, Materials and Methods, and Table 2, the authors stated that group A contains the patients with ulcerative colitis without the presence of ghrelin staining in biopsy samples; whereas patients with ulcerative colitis and presence of ghrelin staining in biopsy samples were gathered in group B. In addition, the authors stated that Mayo score/disease activity index (DAI) is patients with ulcerative colitis was significantly higher in group A than group B (abstract and Table 2). Final conclusions at the end of the manuscript are in agreement with mentioned above data. On the other hand, the authors present opposite conclusion in the abstract that “Colonic ghrelin staining in UC patients seems to be associated with the increased activity of this disease”. Moreover, the authors stated that “DAI were found to be higher in UC patients with showing ghrelin positivity (Group A) (Discussion, paragraph 3). Inconsistences in the conclusions should be checked and corrected.

2. All abbreviations should be presented in full form at the place where they are used for the first time in the abstract and repeated in the main body of the manuscript.

3. Introduction. The authors should write that in the gut, the protective and/or therapeutic effect of ghrelin was demonstrated in different organs (PMID: 34638910), including among others, in the oral cavity (PMID: 24304579; PMID: 29151078), pancreas (PMID: 14726611; PMID: 17084100; PMID: 25594510), stomach (PMID: 24622834) and duodenum (PMID: 19439811).

4. The authors should write that previous studies showed that ghrelin exhibit the beneficial effects in different experimental models of colitis. Among others, the protective and/or therapeutic effect of ghrelin was observed in experimental colitis evoked by acetic acid (PMID: 26769837; PMID: 27598133), dextran sodium sulfate (PMID: 26713317)

5. Major revision of English is necessary.

Reviewer #2: It is not clear how ghrelin is implicated in UC pathogenesis. Why authors choose only UCand no CD patients. Did the patients received biological agents?

It is very simple approach to have only immunohistochemistry data.

What can be the relation of grelin expression and biological therapies

6. PLOS authors have the option to publish the peer review history of their article (what does this mean?). If published, this will include your full peer review and any attached files.

Reviewer #1: No

Reviewer #2: No

---

## [Author Response · Author response to Decision Letter 0]

5 Aug 2022

3.07.2022

Dear Editor,

We would like to thank you for considering our manuscript entitled “ Correlation of tissue ghrelin levels with disease activity and laboratory parameters in ulcerative colitis patients; immunohistochemical study

” (Manuscript No: PONE-D-22-04274 

) for publication in the PLOS ONE. We also would like to thank to the Editorial board and the Referees for their important contributions, which finally improved our paper. 

We have carefully reviewed the comments of the Reviewers and revised the manuscript accordingly. Below please find the answers to the Reviewers’ comments and the revised version of our paper. Please do not hesitate to make any further change in the text according to the requirements of the journal. 

Yours faithfully,

Memduh Sahin, MD

Corresponding author:

Memduh Sahin M.D.

Basaksehir Cam and Sakura City Hospital,

Basaksehir/ İSTANBUL

E-mail:memsahinsahin@otmail.com

The following answers were given and the revisions made according to the Reviewers’ comments:

Reviewer #1 

Comment 1:

“First of all, the authors check correctness of description of groups of people participating in the study and the presented results. In the abstract, Materials and Methods, and Table 2, the authors stated that group A contains the patients with ulcerative colitis without the presence of ghrelin staining in biopsy samples; whereas patients with ulcerative colitis and presence of ghrelin staining in biopsy samples were gathered in group B. In addition, the authors stated that Mayo score/disease activity index (DAI) is patients with ulcerative colitis was significantly higher in group A than group B (abstract and Table 2). Final conclusions at the end of the manuscript are in agreement with mentioned above data. On the other hand, the authors present opposite conclusion in the abstract that “Colonic ghrelin staining in UC patients seems to be associated with the increased activity of this disease”. Moreover, the authors stated that “DAI were found to be higher in UC patients with showing ghrelin positivity (Group A) (Discussion, paragraph 3). Inconsistences in the conclusions should be checked and corrected.

” 

Answer 1:

“We would like to express our gratitude to the Reviewer for this kind appreciation of our work and the outstanding criticisms, which finally improved our paper.’’

In our study, ulcerative colitis disease activity was found to be lower in cases with ghrelin staining. The statements in the abstract and discussion part of the research were corrected with these sentences.

Abstract“. Mayo score/disease activity index (DAI) for UC was significantly higher in Group A compared to Group B (p=0.03). 

 There were no differences in the amount of colonic ghrelin staining between healthy individuals and UC patients. Colonic ghrelin staining in UC patients seemed to be associated with the decreased activity of this disease.’’

Discussionc ‘ ‘In our study, higher levels of staining were found in the control group in terms of tissue staining amounts, but this data was not statistically significant. DAI was found to be higher in UC patients with negative ghrelin (Group A).’’

Conclusion’’Disease activity detected in the group with positive ghrelin staining, was found to be statistically lower than the negative stained group’’

Comment 2:

“All abbreviations should be presented in full form at the place where they are used for the first time in the abstract and repeated in the main body of the manuscript.”

Answer and Revision 2:

Abbreviations in the article are repeated in the same way as they were first mentioned in the whole article. “” .

Comment 3:

“. Introduction. The authors should write that in the gut, the protective and/or therapeutic effect of ghrelin was demonstrated in different organs (PMID: 34638910), including among others, in the oral cavity (PMID: 24304579; PMID: 29151078), pancreas (PMID: 14726611; PMID: 17084100; PMID: 25594510), stomach (PMID: 24622834) and duodenum (PMID: 19439811).

"

Answer and Revision 3: The introduction part has been changed by adding the expressions suggested by the referees. Sources of information on these expressions are also included. “Protective and/or therapeutic effect of ghrelin has been demonstrated in different organs (17), including among others, the oral cavity (18; 19), pancreas (20; 21; 22), stomach (23) and duodenum (24).’’

Comment 4: The authors should write that previous studies showed that ghrelin exhibit the beneficial effects in different experimental models of colitis. Among others, the protective and/or therapeutic effect of ghrelin was observed in experimental colitis evoked by acetic acid (PMID: 26769837; PMID: 27598133), dextran sodium sulfate (PMID: 26713317)

Answer and Revision 4: The introduction part has also been changed by adding the expressions suggested by the referees. Sources of information on these expressions are also included. “Previous studies have shown that ghrelin exhibited the beneficial effects in various experimental models of colitis. Among others, protective and/or therapeutic activity of ghrelin has been observed in experimental colitis induced by acetic acid (25; 26) and dextran sodium sulfate (27).’’

Comment 5: Major revision of English is necessary.

Answer and Revision 5: The article has been re-evaluated in terms of English, and the detected English spelling errors and sentence styles have been changed and corrected.

Reviewer #2

Comment 1: : It is not clear how ghrelin is implicated in UC pathogenesis. Why authors choose only UCand no CD patients. Did the patients received biological agents? It is very simple approach to have only immunohistochemistry data. What can be the relation of grelin expression and biological therapies

Answer and Revision 1: In our study, only ulcerative colitis patients were included in order to make the group homogeneous. A study with Crohn's patients or a comparison of ulcerative colitis and crohn's patient group may be suggestive for another study.

Since only two (2 infliximab treatment patient) patients received biological treatment in the study, statistical evaluation could not be made for this. Our results can be instructive for the comparison of patients who received and did not receive biological treatment. Examination of patients receiving biologic agents may be the subject of another study. The explanations about the mentioned questions and the treatments received by the patients are explained in the discussion section. “In order to ensure homogeneity, only ulcerative colitis patients were included in the study. Studies on Crohn's disease patients may be a prompt for another study. Another limitation in our study was the scarce use of biologic agents in the patient group (only 2 patients were receiving infliximab). Therefore, in the study, treatment with biological agents was evaluated among patients receiving immunosuppressive therapy, and this group includes patients receiving steroid, azathioprine, and infliximab. There were no patients receiving adalimumab treatment in the study.’’

The treatments received by the patients are summarized in Table 4. Immunosuppressive and biological treatments taken by the patients were added to this table in more detail and revision was applied.

Journal Requirements: -During your revisions, please note that a simple title correction is required: 'Relationship' is misspelled. Please ensure this is updated in the manuscript file and the online submission information

- Thank you for stating the following financial disclosure: "No Fınancial Support"

At this time, please address the following queries

b) State what role the funders took in the study. If the funders had no role in your study, please state: “The funders had no role in study design, data collection and analysis, decision to publish, or preparation of the manuscript

Answer and Revision: : The title of the article has been changed to "Correlation between the tissue ghrelin presence, disease activity and laboratory parameters in ulcerative colitis patients: An immunohistochemical study’’

 No financial support was received in the study, and the research costs were covered by the researchers' own budget and means.

---

## [Decision Letter · Decision Letter 1]

18 Aug 2022

PONE-D-22-04274R1Correlation between the tissue ghrelin presence, disease activity and laboratory parameters in ulcerative colitis patients: immunohistochemical studyPLOS ONE

Dear Dr. Sahin,

Thank you for submitting your manuscript to PLOS ONE. After careful consideration, we feel that it has merit but does not fully meet PLOS ONE’s publication criteria as it currently stands. Therefore, we invite you to submit a revised version of the manuscript that addresses the points raised during the review process.

One of the reviewers still raised some comments which should be addressed before sending the final version. Please have in mind that PLOS does not provide proof reading, thus please assure to send your manuscript in proper English. I also agree with the reviewer that an exemplary image showing the staining for ghrelin is desirable; however, it is your decision to present such an image or not.

We look forward to receiving your revised manuscript.

Kind regards,

Gernot Zissel, Ph.D.

Academic Editor

PLOS ONE

Journal Requirements:

Reviewers' comments:

Reviewer's Responses to Questions

**Comments to the Author**

1. If the authors have adequately addressed your comments raised in a previous round of review and you feel that this manuscript is now acceptable for publication, you may indicate that here to bypass the “Comments to the Author” section, enter your conflict of interest statement in the “Confidential to Editor” section, and submit your "Accept" recommendation.

Reviewer #1: (No Response)

Reviewer #2: All comments have been addressed

2. Is the manuscript technically sound, and do the data support the conclusions?

Reviewer #1: Yes

Reviewer #2: Yes

3. Has the statistical analysis been performed appropriately and rigorously? 

Reviewer #1: N/A

Reviewer #2: Yes

4. Have the authors made all data underlying the findings in their manuscript fully available?

Reviewer #1: Yes

Reviewer #2: Yes

5. Is the manuscript presented in an intelligible fashion and written in standard English?

Reviewer #1: No

Reviewer #2: Yes

6. Review Comments to the Author

Reviewer #1: Manuscript ID: AJGASTRO-D-21-00058

Title: Relationship between the tissue ghrelin presence, disease activity and laboratory parameters in ulcerative colitis patients; immunohistochemical study

Authors: Memduh Sahin et al.

The second version of the manuscript exhibits some improvement, but there are still some errors and deficiencies.

List of shortcomings and errors:

1. Authors should proofread English in their manuscript. Even in the newly introduced changes, there are errors. For example, in response to the firs comment, the authors wrote “Disease activity detected in the group with positive ghrelin staining, was found to be statistically lower than the negative stained group”. Between “than” and “the negative stained group”, the authors should add “in”.

2. The second comment was “All abbreviations should be presented in full form at the place where they are used for the first time in the abstract and repeated in the main body of the manuscript”. Authors stated that the manuscript was revised as suggested by the reviewer. However, there are still abbreviations in the abstract and next parts of the manuscript that are not presented in full name. For example, ESP, CRP. Authors should check the entire manuscript. In addition, the authors should specify in which fluids they determined albumin concentration, as well as in Material and Methods, they should present the methodology used in biochemical tests and blood counts.

3. In the previous version of the manuscript, the authors presented exemplary images showing the staining of colonic mucosa for the presence of ghrelin. There are no such images in the current version of the manuscript. They should appear in the final version of the manuscript,

Reviewer #2: the authors covered the reviewer comments. The manuscript has been improved and it can be accepted for publication

7. PLOS authors have the option to publish the peer review history of their article (what does this mean?). If published, this will include your full peer review and any attached files.

Reviewer #1: No

Reviewer #2: No

---

## [Author Response · Author response to Decision Letter 1]

31 Aug 2022

27.08.2022

Dear Editor,

We would like to thank you for considering our manuscript entitled “ Correlation between the tissue ghrelin presence, disease activity and laboratory parameters in ulcerative colitis patients: An immunohistochemical study

” (Manuscript No: PONE-D-22-04274 

) for publication in the PLOS ONE. We also would like to thank to the Editorial board and the Referees for their important contributions, which finally improved our paper. Recent revisions to this letter in addition to previous corrections have been added as subparagraphs.

We have carefully reviewed the comments of the Reviewers and revised the manuscript accordingly. Below please find the answers to the Reviewers’ comments and the revised version of our paper. Please do not hesitate to make any further change in the text according to the requirements of the journal. 

Yours faithfully,

Memduh Sahin, MD

Corresponding author:

Memduh Sahin M.D.

Basaksehir Cam and Sakura City Hospital,

Basaksehir/ İSTANBUL

E-mail:memsahinsahin@otmail.com

The following answers were given and the revisions made according to the Reviewers’ comments:

Reviewer #1 

Comment 1:

“First of all, the authors check correctness of description of groups of people participating in the study and the presented results. In the abstract, Materials and Methods, and Table 2, the authors stated that group A contains the patients with ulcerative colitis without the presence of ghrelin staining in biopsy samples; whereas patients with ulcerative colitis and presence of ghrelin staining in biopsy samples were gathered in group B. In addition, the authors stated that Mayo score/disease activity index (DAI) is patients with ulcerative colitis was significantly higher in group A than group B (abstract and Table 2). Final conclusions at the end of the manuscript are in agreement with mentioned above data. On the other hand, the authors present opposite conclusion in the abstract that “Colonic ghrelin staining in UC patients seems to be associated with the increased activity of this disease”. Moreover, the authors stated that “DAI were found to be higher in UC patients with showing ghrelin positivity (Group A) (Discussion, paragraph 3). Inconsistences in the conclusions should be checked and corrected.

” 

Answer 1:

“We would like to express our gratitude to the Reviewer for this kind appreciation of our work and the outstanding criticisms, which finally improved our paper.’’

In our study, ulcerative colitis disease activity was found to be lower in cases with ghrelin staining. The statements in the abstract and discussion part of the research were corrected with these sentences.

Abstract“. Mayo score/disease activity index (DAI) for UC was significantly higher in Group A compared to Group B (p=0.03). 

 There were no differences in the amount of colonic ghrelin staining between healthy individuals and UC patients. Colonic ghrelin staining in UC patients seemed to be associated with the decreased activity of this disease.’’

Discussionc ‘ ‘In our study, higher levels of staining were found in the control group in terms of tissue staining amounts, but this data was not statistically significant. DAI was found to be higher in UC patients with negative ghrelin (Group A).’’

Conclusion’’Disease activity detected in the group with positive ghrelin staining, was found to be statistically lower than the negative stained group’’

Comment 2:

“All abbreviations should be presented in full form at the place where they are used for the first time in the abstract and repeated in the main body of the manuscript.”

Answer and Revision 2:

Abbreviations in the article are repeated in the same way as they were first mentioned in the whole article. “” .

Comment 3:

“. Introduction. The authors should write that in the gut, the protective and/or therapeutic effect of ghrelin was demonstrated in different organs (PMID: 34638910), including among others, in the oral cavity (PMID: 24304579; PMID: 29151078), pancreas (PMID: 14726611; PMID: 17084100; PMID: 25594510), stomach (PMID: 24622834) and duodenum (PMID: 19439811).

"

Answer and Revision 3: The introduction part has been changed by adding the expressions suggested by the referees. Sources of information on these expressions are also included. “Protective and/or therapeutic effect of ghrelin has been demonstrated in different organs (17), including among others, the oral cavity (18; 19), pancreas (20; 21; 22), stomach (23) and duodenum (24).’’

Comment 4: The authors should write that previous studies showed that ghrelin exhibit the beneficial effects in different experimental models of colitis. Among others, the protective and/or therapeutic effect of ghrelin was observed in experimental colitis evoked by acetic acid (PMID: 26769837; PMID: 27598133), dextran sodium sulfate (PMID: 26713317)

Answer and Revision 4: The introduction part has also been changed by adding the expressions suggested by the referees. Sources of information on these expressions are also included. “Previous studies have shown that ghrelin exhibited the beneficial effects in various experimental models of colitis. Among others, protective and/or therapeutic activity of ghrelin has been observed in experimental colitis induced by acetic acid (25; 26) and dextran sodium sulfate (27).’’

Comment 5: Major revision of English is necessary.

Answer and Revision 5: The article has been re-evaluated in terms of English, and the detected English spelling errors and sentence styles have been changed and corrected.

Reviewer #2

Comment 1: : It is not clear how ghrelin is implicated in UC pathogenesis. Why authors choose only UCand no CD patients. Did the patients received biological agents? It is very simple approach to have only immunohistochemistry data. What can be the relation of grelin expression and biological therapies

Answer and Revision 1: In our study, only ulcerative colitis patients were included in order to make the group homogeneous. A study with Crohn's patients or a comparison of ulcerative colitis and crohn's patient group may be suggestive for another study.

Since only two (2 infliximab treatment patient) patients received biological treatment in the study, statistical evaluation could not be made for this. Our results can be instructive for the comparison of patients who received and did not receive biological treatment. Examination of patients receiving biologic agents may be the subject of another study. The explanations about the mentioned questions and the treatments received by the patients are explained in the discussion section. “In order to ensure homogeneity, only ulcerative colitis patients were included in the study. Studies on Crohn's disease patients may be a prompt for another study. Another limitation in our study was the scarce use of biologic agents in the patient group (only 2 patients were receiving infliximab). Therefore, in the study, treatment with biological agents was evaluated among patients receiving immunosuppressive therapy, and this group includes patients receiving steroid, azathioprine, and infliximab. There were no patients receiving adalimumab treatment in the study.’’

The treatments received by the patients are summarized in Table 4. Immunosuppressive and biological treatments taken by the patients were added to this table in more detail and revision was applied.

Journal Requirements: -During your revisions, please note that a simple title correction is required: 'Relationship' is misspelled. Please ensure this is updated in the manuscript file and the online submission information

- Thank you for stating the following financial disclosure: "No Fınancial Support"

At this time, please address the following queries

b) State what role the funders took in the study. If the funders had no role in your study, please state: “The funders had no role in study design, data collection and analysis, decision to publish, or preparation of the manuscript

Answer and Revision: : The title of the article has been changed to "Correlation between the tissue ghrelin presence, disease activity and laboratory parameters in ulcerative colitis patients: An immunohistochemical study’’

 No financial support was received in the study, and the research costs were covered by the researchers' own budget and means.

Secondary Revisions

Comment 1: Authors should proofread English in their manuscript. Even in the newly introduced changes, there are errors. For example, in response to the first comment, the authors wrote “Disease activity detected in the group with positive ghrelin staining, was found to be statistically lower than the negative stained group”. Between “than” and “the negative stained group”, the authors should add “in”.

Answer and Revision 1: In addition to the sentence stated by the esteemed referee, the entire article was reviewed by a professional translator as English errors and relevant corrections were made in this regard. Improvements made are detailed in the Tracked Changes document.

Comment 2: The second comment was “All abbreviations should be presented in full form at the place where they are used for the first time in the abstract and repeated in the main body of the manuscript”. Authors stated that the manuscript was revised as suggested by the reviewer. However, there are still abbreviations in the abstract and next parts of the manuscript that are not presented in full name. For example, ESP, CRP. Authors should check the entire manuscript. In addition, the authors should specify in which fluids they determined albumin concentration, as well as in Material and Methods, they should present the methodology used in biochemical tests and blood counts.

Answer and Revision 2: All abbreviations that are requested to be corrected and explained are added to the article with their explanations and abbreviations. All corrections made can be seen in detail in the Tracked Changes File. In our research, Albumin Levels were evaluated in serum samples and sentences on this subject were added to the article. In addition, a short paragraph has been added about the blood analyzes performed in our research.’’Blood Serum and CBC (Complet Blood Cell Count) samples were taken from the antecubital area of the subjects between 7 a.m. and 9 a.m., after 8 h fasting. Hemoglobin, ESR, White Blood Cell count, serum albumin and CRP values of all subjects (UC patients and control group) were analyzed. CBC samples installed in Ethylene Diamine Tetra Acetic Acid (EDTA) tubes. The Serum samples were collected in clean polypropylene tube ( Blood centrifuged at 3000 rpm for 15 min).’’

Comment 3: In the previous version of the manuscript, the authors presented exemplary images showing the staining of colonic mucosa for the presence of ghrelin. There are no such images in the current version of the manuscript. They should appear in the final version of the manuscript,

Answer and Revision 2: Figures previously added to the article were loaded into the figure system as Figure 1 and Figure 2.

---

## [Editor Report · Decision Letter 2]

28 Sep 2022

Correlation between the tissue ghrelin presence, disease activity and laboratory parameters in ulcerative colitis patients: immunohistochemical study

PONE-D-22-04274R2

Dear Dr. Sahin,

We’re pleased to inform you that your manuscript has been judged scientifically suitable for publication and will be formally accepted for publication once it meets all outstanding technical requirements.

Kind regards,

Gernot Zissel, Ph.D.

Academic Editor

PLOS ONE
---

## [Editor Report · Acceptance letter]

14 Oct 2022

PONE-D-22-04274R2 

Correlation between the tissue ghrelin presence, disease activity and laboratory parameters in ulcerative colitis patients; immunohistochemical study 

Dear Dr. Sahin:

I'm pleased to inform you that your manuscript has been deemed suitable for publication in PLOS ONE. Congratulations! Your manuscript is now with our production department. 

Kind regards, 

on behalf of

Prof. Dr. Gernot Zissel 

Academic Editor

PLOS ONE